

# From data to insights: a tool for comprehensive Quantification of Continuous Glucose Monitoring (QoCGM)

Simon Cichosz[1], Stine Hangaard[1,2], Thomas Kronborg[1,2], Peter Vestergaard[2,3] and Morten Hasselstrøm Jensen[1,4]

[1] Department of Health Science and Technology, Aalborg University, Gistrup, Denmark
[2] Steno Diabetes Center North Denmark, Aalborg University Hospital, Aalborg, Denmark
[3] Department of Endocrinology, Aalborg University Hospital, Aalborg, Denmark
[4] Data Science, Novo Nordisk, Soeborg, Denmark

## ABSTRACT

Continuous glucose monitoring (CGM) has become a important technology in the management and research of both type 1 and type 2 diabetes, providing real-time data on glucose fluctuations that were previously inaccessible with traditional monitoring methods. Numerous analytical tools have been developed for platforms like R and Python to calculate standard metrics and extract insights from CGM data. However, these tools often fail to address the full spectrum of analytical requirements. Furthermore, there is a lack of updated, open-source tools tailored for MATrix LABoratory (MATLAB)—a platform widely used by the research community. To address this gap, we introduce Quantification of Continuous Glucose Monitoring (QoCGM), a comprehensive, open-source post-hoc analytical tool for CGM data specifically designed for MATLAB. A case study involving 324 individuals with insulin-treated type 2 diabetes mellitus (T2DM) demonstrates the utility of QoCGM, highlighting the distinct aspects of glucose dynamics captured by different CGM-derived metrics through an analysis of their coefficients of determination ($R^2$).

## INTRODUCTION

Continuous glucose monitoring (CGM) has emerged as a cornerstone in the management and study of both type 1 and type 2 diabetes (*The Juvenile Diabetes Research Foundation Continuous Glucose Monitoring Study Group, 2008*; *Rodbard, 2016*; *Carlson, Mullen & Bergenstal, 2017*; *Martens et al., 2021*), offering real-time insights into glucose fluctuations that were previously unattainable through traditional monitoring methods. The data generated by CGM devices provides an unprecedented opportunity for both clinical and scientific applications, enabling precise glycemic control, better prediction of hypoglycemic and hyperglycemic events, and a deeper understanding of glucose dynamics. Key metrics

Corresponding author
Simon Cichosz, simcich@hst.aau.dk

derived from CGM data, such as time-in-ranges and glycemic variability have become integral to both patient care and diabetes research, facilitating personalized treatment plans and the development of novel therapeutic strategies (*Danne et al., 2017*; *Battelino et al., 2019*; *Battelino et al., 2023*).

In several previous studies, we have shown how CGM derived metrics can be utilized in the prediction of emerging hypoglycemic events (*Cichosz et al., 2014*; *Fleischer, Hansen & Cichosz, 2022*; *Thomsen et al., 2023*; *Kronborg et al., 2024*), week-to-week risk of excessive hyperglycemia, hypoglycemia and glycemic variability (*Cichosz, Jensen & Olesen, 2024*), elevated ketone levels, (*Cichosz & Bender, 2024*) and identification of gastroparesis (*Cichosz & Hejlesen, 2022*).

Over the years, several analytical tools have been developed to harness the potential of CGM data, providing capabilities for calculating standard metrics and generating insights from the complex glucose profiles captured (*Piersanti et al., 2023*; *Olsen et al., 2024*). Examples of such tool are cgmquantify, iglu, GLU, rGV, and CGManalyzer, which are all available for R or Python (*Millard et al., 2020*; *Bent, Henriquez & Dunn, 2021*; *Broll et al., 2021*; *Olawsky et al., 2022*). Also, the AGATA toolbox exists for analytics needs in MATLAB/ Octave environment and is focused on visualization of the analytic results (*Cappon, Sparacino & Facchinetti, 2024*). However, these tools do often not address the full spectrum of analytical needs. Specifically, there is a growing demand for tools that can incorporate newer, clinically relevant metrics, perform nuanced analyses of nocturnal *versus* diurnal glucose patterns, event detection of hypoglycemia, novel glycemic variability metrics and assess day-to-day variations in glucose levels—factors that are critical for both advanced research and personalized clinical interventions. Moreover, while various software solutions exist, there is a notable gap in the availability of updated, open-source tools for the MATLAB environment (*Olsen et al., 2024*), which is widely used in the Engineering research community.

This article presents an open-source tool Quantification of Continuous Glucose Monitoring (QoCGM) designed for comprehensive *post-hoc* CGM data analysis within the MATLAB environment. This tool does not only include newer metrics but also offers features for examining night/day glucose variations and day-to-day changes, making it a valuable resource for researchers and clinicians. Portions of this text were previously published as part of a preprint (https://doi.org/10.1101/2025.01.01.25319870).

## IMPLEMENTATION

QoCGM is implemented in MATLAB and has been tested with version R2021b. The source code is available on GitHub (https://github.com/simcich/QoCGM, DOI: 10.5281/zenodo.15018569). QoCGM is designed to process preprocessed CGM data from comma-separated value (CSV) files. The input CSV files should contain two columns: one for the timestamp in the format 'YYYY-MM-DD HH:MM:SS', and one for the corresponding glucose values, which can be in either mg/dL or mmol/L.

The processing of CGM data in QoCGM occurs in two primary stages: (i) preprocessing and handling missing data, and (ii) deriving metrics. The source code includes examples
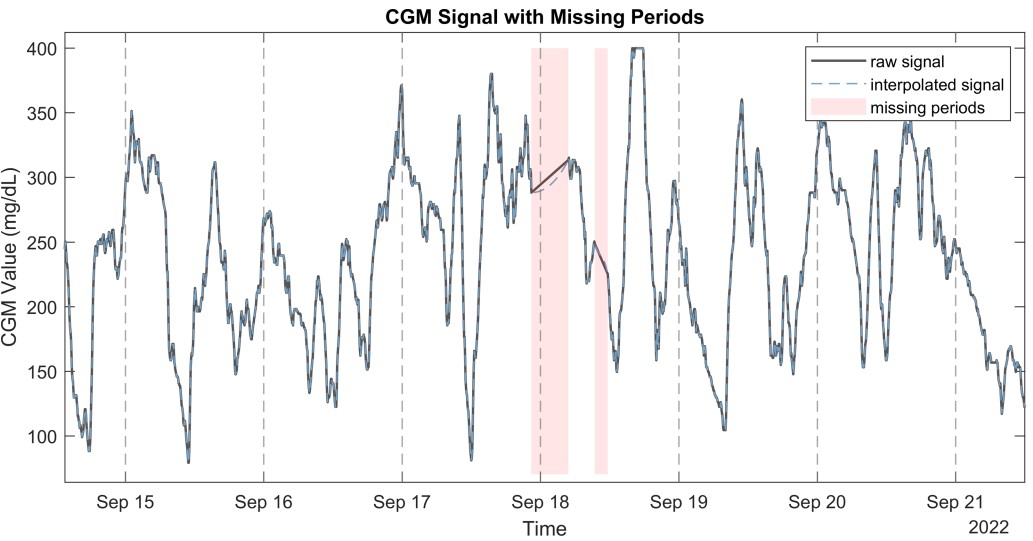

**Figure 1** Raw CGM signal with the corresponding interpolated signal and missing periods.

that demonstrate how to use QoCGM with a single file or to batch process multiple files within a directory.

To run QoCGM, users load the CGM data into a table and provide the following additional arguments:

- Sampling frequency: A numeric value representing the expected sampling frequency in minutes. For example, set this to 5 if measurements are expected every 5 min.
- Sampling tolerance: A numeric threshold (in minutes) defining the acceptable variation in sampling times. Measurements that fall outside this threshold are corrected.
- End of night: An integer (0–24) specifying the hour at which the 'morning' period begins, thereby marking the end of the 'night'. For instance, set this to 6 for a 6 AM start.
- Conversion flag: A binary flag (1 or 0) indicating whether to convert CGM values from mmol/L to mg/dL. Since the metrics are derived based on mg/dL values, conversion is necessary if the CGM units are in mmol/L.
- Plotting flag: A binary flag, set to 1 to enable plotting of figures related to signal quality control and data processing. Examples of these plots are presented in Figs. 1–3.
- Handling of missing data: either interpolation (default) or by removing periods with missing data.

Upon execution, QoCGM generates a CSV file containing the derived CGM metrics, which can then be imported into statistical software for further analysis.

## Preprocessing and handling missing data
### Data preprocessing and removal of duplicates
The preprocessing of the CGM signal begins with the removal of duplicate entries in the dataset. The process removes duplicate values that share identical timestamps, as these can arise during data export from CGM platforms or due to synchronization issues. This step
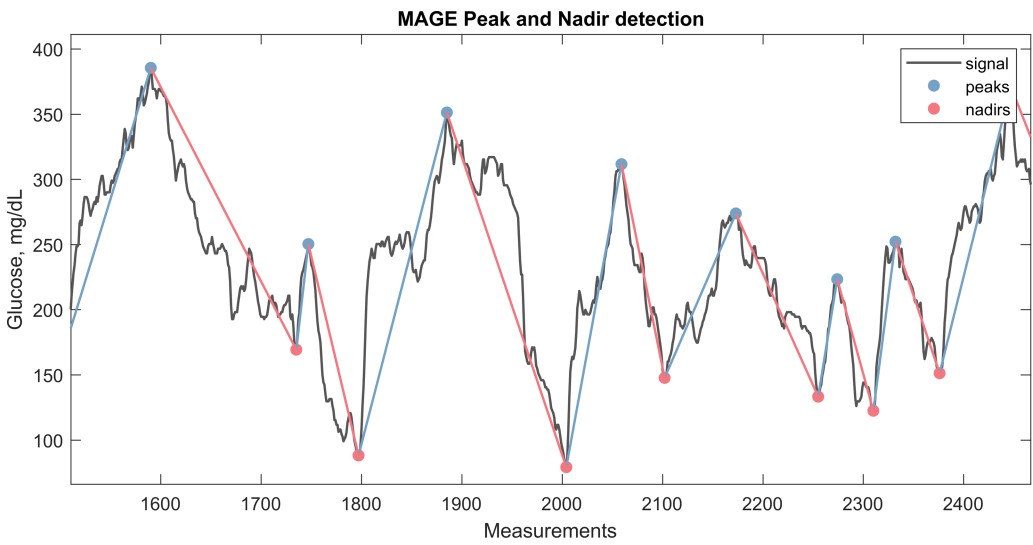

**Figure 2   CGM signal with the corresponding nadirs and peaks detected for calculation of MAGE.** The plot illustrate a window from approximately 3 days of CGM monitoring.

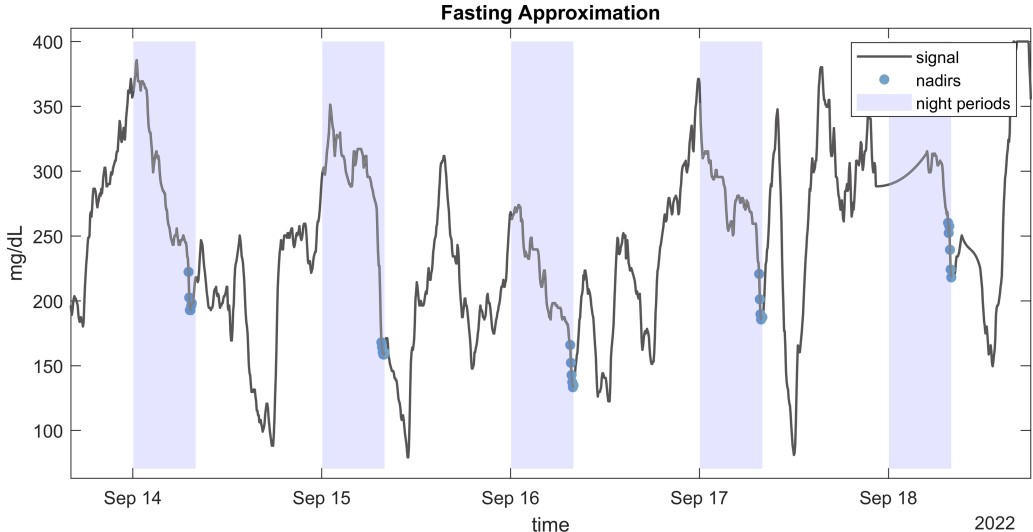

**Figure 3   Fasting approximation.** Example of CGM signal with highlighted detected nadirs values during nights used for fasting approximation (FGxP).

is critical to ensure that each time point in the data is unique, preventing potential errors in subsequent analyses.

### Identification of missing periods in the CGM signal

CGM systems are configured to sample glucose levels at specific intervals. For example, a common sampling frequency involves recording data every 5 min. This interval is considered the expected time between consecutive measurements.

Small deviations from the expected sampling interval can occur due to natural variability in device performance or other factors. To account for this, a tolerance range is established around the expected interval, defining the maximum allowable intervals between consecutive readings. The time differences between consecutive glucose measurements are calculated to identify intervals that fall outside the permissible % range. Any interval that is longer than the defined tolerance indicates a potential gap or missing period in the data. The specific points in time where data is missing are identified by examining the calculated time differences. These points mark the beginning of periods when the expected glucose measurements are absent.

### Interpolation of missing data

To reconstruct the missing portions of the CGM signal, interpolation is performed over the identified gaps using Piecewise Cubic Hermite Interpolating Polynomial (PCHIP) (*Rabbath & Corriveau, 2019*).

The final PCHIP interpolating function H(x) is defined piecewise over each interval $(x_i, x_{i+1})$ as:

$$H(x) = H_i(x) \ for \ x_i \leq x \leq x_{i+1}.$$

This method was chosen for its ability to maintain the shape and smoothness of the original data, making it suitable for CGM signals, which typically exhibit smooth and continuous changes in glucose levels. This reconstructed signal filled the gaps identified earlier, providing a continuous and complete signal for further analysis. Figure 1 illustrates a CGM signal with missing periods, the raw signal, and the PCHIP interpolation. It is also possible to handle missing periods without interpolation by setting input argument to removing these periods.

## CGM derived metrics

After preprocessing and interpolation of the signal, QoCGM derives a set of CGM metrics, listed in Table 1. Glucose metrics output by QoCGM were chosen to represent categories of glucose characteristics that reflect a set of broad domains that relate to outcomes, glycemic control, or be influenced by exposures in clinical trials.

The domains are:

- Basic descriptive statistics
- Time-in-range (TIR) metrics
- Glycemic risk indicators
- Glycemic variability metrics
- Glycemic control indicators
- Entropy and complexity measures

The QoCGM tool calculates basic descriptive statistics and time-in-range (TIR) metrics for the entire day, as well as separately for nighttime and daytime periods. Additionally, it provides the day-to-day standard deviation for selected variables. All other metrics are derived from the complete CGM signal provided.
**Table 1** The figure displays the CGM metrics calculated by QoCGM with a short description.

| Metric | Description | All | Day | Night |
|---|---|---|---|---|
| MonitoringDays | Number of days with monitoring | X | | |
| CompletenessRate | The completeness of the monitored period | X | | |
| sample_n | Number of samples | X | X | X |
| Mean | Mean glucose value | X | X | X |
| Median | Median glucose value | X | X | X |
| Std | Standard deviation of glucose values | X | X | X |
| CV | Coefficient of variation | X | X | X |
| IQR | Interquartile range | X | X | X |
| Pctile75 | 75th percentile of glucose values | X | X | X |
| Pctile25 | 25th percentile of glucose values | X | X | X |
| TIR | Time in range (70–180 mg/dL) | X | X | X |
| TITR | Time in tight range (70–140 mg/dL) | X | X | X |
| TBR1 | Time below range (54–70 mg/dL) | X | X | X |
| TBR2 | Time below range (<54 mg/dL) | X | X | X |
| TBR | Total time below range (TBR1 + TBR2) | X | X | X |
| TAR1 | Time above range (180–250 mg/dL) | X | X | X |
| TAR2 | Time above range (>250 mg/dL) | X | X | X |
| TAR | Total time above range (TAR1 + TAR2) | X | X | X |
| Hypo_episodes_n | Number of hypoglycemia events (<70 mg/dL) | X | | |
| GRI_Hypo | Glucose Risk Index for hypoglycemia | X | | |
| GRI_Hyper | Glucose Risk Index for hyperglycemia | X | | |
| GRI | Glucose Risk Index | X | | |
| CONGA_1H | Continuous Overall Net Glycemic Action over 1 h | X | | |
| CONGA_2H | Continuous Overall Net Glycemic Action over 2 h | X | | |
| CONGA_6H | Continuous Overall Net Glycemic Action over 6 h | X | | |
| CONGA_24H | Continuous Overall Net Glycemic Action over 24 h | X | | |
| MAGE | Mean Amplitude of Glycemic Excursion | X | | |
| Mobility | Signal mobility | X | | |
| DTpM | Distance traveled per minute | X | | |
| FGxP | Fasting glucose proxy | X | | |
| GMI | Glucose Management Indicator | X | | |
| LBGI | Low Blood Glucose Index | X | | |
| HBGI | High Blood Glucose Index | X | | |
| MCI | Multiscale Complexity Index | X | | |
| GRADE | Glycemic Risk Assessment Diabetes Equation score | X | | |
| GRADE_hypo | Percentage of GRADE score for hypoglycemia | X | | |
| GRADE_eu | Percentage of GRADE score for euglycemia | X | | |
| GRADE_hyper | Percentage of GRADE score for hyperglycemia | X | | |
| D2d_mean | Day-to-day standard deviation of mean glucose | X | | |
| D2d_TIR | Day-to-day standard deviation of time in range | X | | |

QoCGM aggregates the average of each summary variable across all days for each participant, resulting in a single overall value for each metric. For instance, QoCGM reports TIR metrics such as: (i) mean TIR for whole days, (ii) TIR for nighttime based on a predefined interval, (iii) TIR for daytime based on a predefined interval, and (iv) day-to-day variation in TIR. Metrics for each day for a given individual can be calculated by splitting the CGM signal into separate files corresponding to unique days.

### Basic descriptive statistics

We included a range of basic descriptive statistics to characterize the CGM data for each participant. These metrics provide foundational insights into the central tendency, dispersion, and distribution of glucose values over the monitored period.

- Mean glucose: The average glucose level over the monitoring period.
- Median glucose: The midpoint of the glucose distribution, less influenced by outliers.
- Standard deviation (SD): A measure of glucose variability around the mean.
- Coefficient of variation (CV): The standard deviation normalized by the mean, expressed as a percentage.
- Interquartile range (IQR): The range between the 25th and 75th percentiles, representing the spread of the middle 50% of glucose values.
- 75th and 25th percentiles: Indicators of the upper and lower bounds of the middle half of the glucose distribution.

These metrics were calculated for the entire period, as well as separately for daytime and nighttime, to capture diurnal variations in glucose levels.

### Time-in-range (TIR) metrics

TIR metrics were employed to assess the proportion of time that glucose levels remained within clinically relevant thresholds, providing a detailed view of glycemic control. The following TIR metrics were included in the analysis:

- TIR (70–180 mg/dL) (*Danne et al., 2017*): Percentage of time within the target range, indicating overall glycemic control.
- TITR (70–140 mg/dL) (*Beck et al., 2024*): Percentage of time within a narrower target range, reflecting tighter glucose control.
- TBR1 (54–70 mg/dL) (*Danne et al., 2017*): Percentage of time in level 1 hypoglycemia.
- TBR2 (<54 mg/dL) (*Danne et al., 2017*): Percentage of time in level 2 hypoglycemia.
- Total TBR (*Hill et al., 2007*): Combined percentage of time in TBR1 and TBR2.
- TAR1 (180–250 mg/dL) (*Danne et al., 2017*): Percentage of time in level 1 hyperglycemia.
- TAR2 (>250 mg/dL) (*Danne et al., 2017*): Percentage of time in level 2 hyperglycemia.
- Total TAR (*Danne et al., 2017*): Combined percentage of time in TAR1 and TAR2.

These metrics were calculated for the whole day and separately for daytime and nighttime.

### Glycemic risk indicators

Glycemic risk indicators were calculated to quantify the risk associated with hyperglycemia and hypoglycemia, providing a nuanced assessment of the overall glycemic profile. The

following metrics were included from the Glycemic Risk Index (GRI), Low/High Blood Glucose Index (LBGI/HBGI), and The Glycemic Risk Assessment in Diabetes (GRADE):

- Glycemic Risk Index for Hypoglycemia (GRI Hypo) (*Klonoff et al., 2023*): This metric quantifies the risk associated with hypoglycemic episodes, with higher values indicating a greater risk of low glucose levels.
- Glycemic Risk Index for Hyperglycemia (GRI Hyper) (*Klonoff et al., 2023*): This metric assesses the risk of hyperglycemia, with higher values indicating a greater risk of elevated glucose levels.
- Overall Glycemic Risk Index (GRI) (*Klonoff et al., 2023*): A composite metric combining the risks of both hypoglycemia and hyperglycemia, calculated as a weighted sum of GRI Hypo and GRI Hyper.
- Low Blood Glucose Index (LBGI) (*Kovatchev et al., 1998*): This score measures the risk of experiencing low blood glucose levels, with higher values indicating increased hypoglycemia risk.
- High Blood Glucose Index (HBGI) (*Kovatchev et al., 2002*): This score quantifies the risk of high blood glucose levels, with higher values indicating an increased risk of hyperglycemia.
- GRADE (Overall) (*Hill et al., 2007*): This metric provides a summary measure of glycemic risk across all glucose ranges, combining the risks of hypoglycemia, euglycemia, and hyperglycemia.
- GRADE for hypoglycemia (*Hill et al., 2007*): The specific component of the GRADE metric that assesses the risk of hypoglycemia.
- GRADE for euglycemia (*Hill et al., 2007*): The component of the GRADE metric that evaluates the time spent in the euglycemic range (normal glucose levels).
- GRADE for hyperglycemia (*Hill et al., 2007*): The component of the GRADE metric that quantifies the risk associated with hyperglycemia.

These glycemic risk indicators were calculated based on the entire CGM signal, offering a detailed view of the participant's glycemic risk profile. A detailed mathematical definition of the metrics is provided in Supplementary Material S1.

### Glycemic variability metrics

Glycemic Variability Metrics were utilized to assess fluctuations in glucose levels over time, providing insights into the stability and predictability of glycemic control. The following metrics were included:

- Continuous overall net glycemic action (CONGA) (*Mcdonnell et al., 2005*): This metric measures glucose variability over different time intervals. We calculated CONGA for 1-hour, 2-hour, 6-hour, and 24-hour periods (CONGA 1H, CONGA 2H, CONGA 6H, and CONGA 24H), reflecting short-term to daily fluctuations in glucose levels.
- Mean amplitude of glycemic excursions (MAGE) (*Service et al., 1970*): MAGE captures the average magnitude of significant glucose swings, both increases and decreases, by focusing on excursions that exceed one standard deviation from the mean. It is a widely used indicator of glycemic variability and the likelihood of large glucose fluctuations.

- Mobility: This metric assesses the signal mobility by measuring the variance of glucose changes (differences between consecutive glucose readings) relative to the overall variance of the glucose signal. Higher mobility indicates greater variability in glucose levels. The metrics have especially been shown to separate healthy form individuals with dysglycemia (*Cichosz et al., 2025*)
- Distance traveled per minute (DTpM) (*Peyser et al., 2018*): DTpM quantifies the total amount of glucose fluctuation over time by summing the absolute changes in glucose levels and normalizing by the total duration of monitoring. This metric provides a rate of glucose change, highlighting periods of rapid fluctuations. The metric is closely related to the established metric mean absolute glucose (MAG) (*Hermanides et al., 2010*; *Kohnert et al., 2013*)
- Standard deviation day-to-day (SD d2d): This metric measures the day-to-day variability in glucose levels, specifically assessing the mean glucose and TIR variability across days. It captures the consistency of glycemic control from one day to the next.

These glycemic variability metrics provide a comprehensive understanding of the dynamics of glucose levels, identifying periods of instability that may increase the risk of hypo- or hyperglycemic events and contribute to long-term complications in diabetes management. A detailed mathematical definition of the metrics is provided in Supplementary Material S1. For example, in Fig. 2, the peaks and nadirs above $\sigma$ (1 SD) is highlighted in the CGM trace. The absolute excursions $|\lambda|$, both positive and negatives, are used to calculate the MAGE metric as show in the equation below:

$$MAGE = \frac{1}{n}\sum_{i=1}^{n}|\lambda_i|\,if\,(|\lambda_i| \geq \sigma).$$

### Glycemic control indicators

Glycemic control indicators were assessed to evaluate the effectiveness of diabetes management strategies by examining specific aspects of glucose control. The following metrics were included in the analysis:

- Fasting glucose proxy (FGxP) (*Millard et al., 2020*): This metric estimates the fasting glucose levels based on the CGM data. It serves as a proxy for assessing baseline glycemic control and helps to approximate the glucose levels typically observed during fasting periods.
- Glucose management indicator (GMI) (*Bergenstal et al., 2018*): The GMI provides an estimate of average glucose levels by applying a formula that correlates with hemoglobin A1C values. This metric offers a standardized measure of overall glycemic control and helps in evaluating the long-term efficacy of diabetes management.
- Hypoglycemia events: This metric quantify the numbers of hypoglycemic episodes below 70 mg/dL for a minimum duration of 15 min.

Several methodologies have been proposed for approximating fasting glucose levels from CGM data. We employed a robust approach developed by *Millard et al. (2020)* which does not require information about mealtimes. This method estimates fasting glucose

by calculating the mean of the 30 lowest consecutive minutes of glucose readings during nighttime. Figure 3 illustrates the CGM readings used to estimate fasting glucose from an individual.

GMI is calculated based on *Bergenstal et al. (2018)*:

$$GMI\ (\%) = 3.31 + 0.02392 \times Mean\ Glucose\ [mg/dL].$$

### *Entropy and complexity measures*

The entropy and complexity measure multiscale entropy (MSE) were utilized to evaluate the irregularity and unpredictability of glucose fluctuations, providing insights into the dynamic nature of glycemic control. MSE quantifies the complexity of the glucose time series by analyzing its entropy across multiple time scales. This metric captures the degree of randomness and structure within glucose data, providing a comprehensive measure of variability and the underlying complexity of glycemic fluctuations. The calculation involves embedding the data into various time scales and assessing the entropy at each scale. The sum of these entropy measures gives the Multiscale Entropy index, reflecting the overall complexity of the glucose signal. We adopted the implementation described by *Kohnert et al. (2017)* where Multiscale Complexity Index (MCI) is defined as the sum over the range of scales, from 1 to 7, at the window length $m = 2$, the sensitivity criterion $r = 0.15$ times the standard deviation. This metric has previously been as an early marker for changes in glucose control (*Colás et al., 2019*).

Tip: To reduce computational time significantly, the calculation of this metric can be omitted by removing the MSE code part.

### *Computation time*

The average computation time for a 30-day CGM signal sampled at 288 points per day is approximately 24.3 s for the full set of metrics, and 4.6 s when excluding MSE—which is notably the most computationally intensive metric. These benchmarks were obtained on a PC with an 11th Gen Intel(R) Core(TM) i7-11850H @ 2.50 GHz processor and 32 GB of RAM.

## Example of usage

In this section, we demonstrate the application of QoCGM by deriving metrics from CGM data collected from individuals with Type 2 diabetes mellitus (T2DM) undergoing insulin therapy. We investigate the correlations between the various metrics generated by QoCGM.

### *Study sample*

We used CGM data (Dexcom, G6) from the *Diabetes teleMonitoring of patients in insulin Therapy (DiaMonT) trial (NCT04981808)*, comprising 331 people with insulin-treated T2DM (*Hangaard et al., 2022*; *Hangaard et al., 2025*). We encompassed participants from both the intervention and the control arm. Written and oral informed consent was obtained from each participant prior to their enrollment in the study. For consistency, we limited the analysis to the initial seven days of CGM monitoring post the inclusion date for all participants. We included all participants with any measurements in the first seven days.

The baseline characteristics of the participants were as follows: mean age 61.3 years (SD 10.6), mean duration of diabetes 17.5 years (SD 11.5), mean BMI 33.1 kg/m$^2$ (SD 6.4), mean HbA1c 64.0 mmol/mol (SD 14.4), and 61.6% of the participants were male. QoCGM was used to calculate CGM metric based on the seven-day profile from each participant. Sampling frequency was set to 5 min, with a sampling variation allowed threshold of 0.05, morning start was set to 8AM, and PCHIP interpolation was applied on periods without CGM coverage.

### Analyses

We conducted a coefficient of determination analysis ($R^2$) to explore the linear relationships between the derived CGM metrics. $R^2$ represents the proportion of variance in one metric that is explained by another metric. $R^2$ ranges from 0 to 1, where 1 indicates that the variability in one variable can be completely explained by the other, and 0 indicates no explanatory power. This analysis provides insights into how different CGM-derived metrics are interrelated, offering guidance on their potential differences.

## RESULTS

A total of 324 participants with T2DM had continuous glucose monitoring (CGM) data available for the first week following enrollment.

The median (25th; 75th percentile) glucose level of the participants was 162 mg/dL (145; 193 mg/dL), with a time-in-range (TIR) of 65% (39; 78%), and a coefficient of variation (CV) of 24% (21; 29%). The median CGM coverage was 100% (99; 100%). The full summery statistics for the full period, night and daytime can be found in Supplementary Material S2.

The coefficient of determination ($R^2$) analysis is illustrated in a heatmap, Fig. 4. Furthermore, coefficient of determination matrix plots with annotation of value is provided in Supplementary Material S2.

This analysis indicates that many metrics exhibit significant overlap with one or more other metrics. Notably, a few metrics from the whole-signal category show a low $R^2$ value (<0.5) when compared with other metrics. These include the day-to-day standard deviation of TIR and mean, mean continuous improvement (MCI), Mobility, and DTpM. Figure 5 illustrates the six metrics with the lowest cumulative $R^2$ values across all metrics. For each of these metrics, the corresponding highest $R^2$ values are depicted in a circular plot.

The analysis of diurnal and nocturnal metrics (Supplementary Material S2) also highlights significant differences between TIR metrics calculated from diurnal and nocturnal periods, underscoring the importance of measuring these as distinct entities in scientific studies on glucose regulation. QoCGM supports this approach by enabling the calculation of metrics for both nighttime and daytime periods, in addition to metrics for the full CGM signal.

## CONCLUSIONS

In this paper, we introduced QoCGM, an open-source tool designed for researchers working with CGM data. QoCGM automates the preprocessing of data and derives a
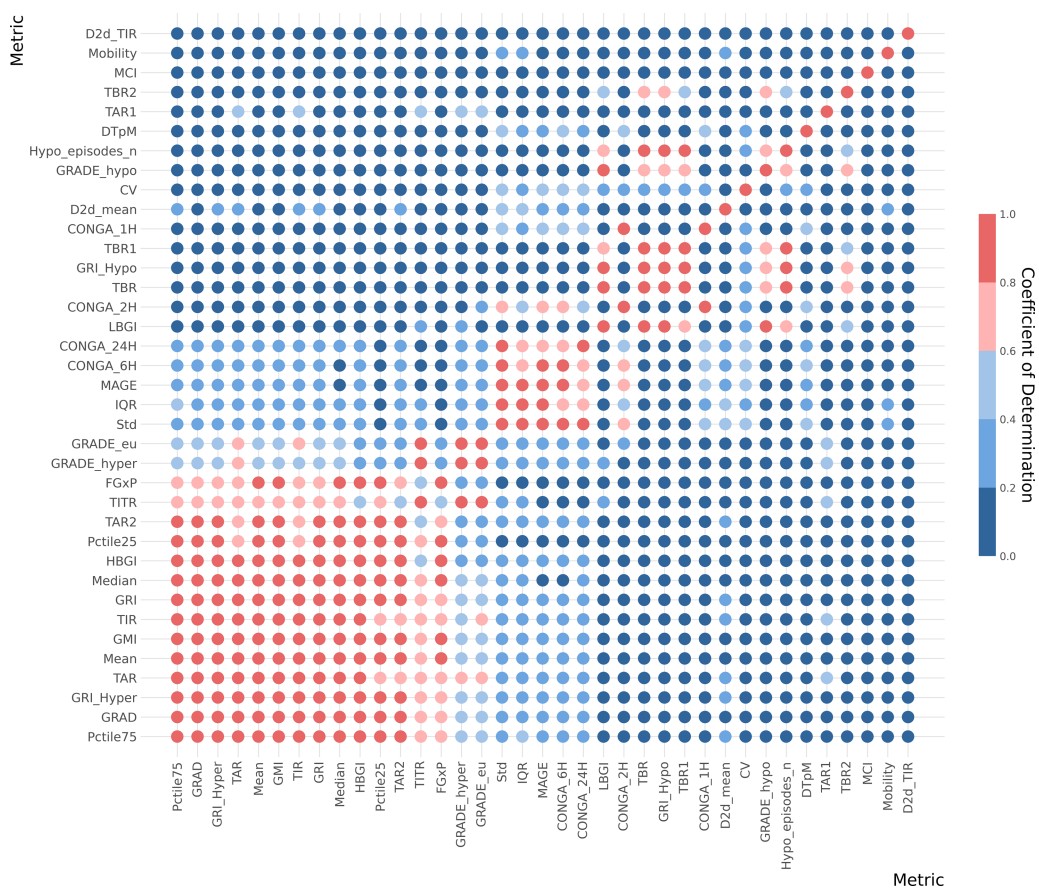

**Figure 4** **Correlation heatmap for whole data CGM metrics.** Metrics is arranged (top to bottom) with the lowest cumulative $R^2$ values across all metrics at top and the highest cumulative $R^2$ values at the bottom of the y-axe. The five color gradient from dark blue to dark red show the $R^2$ tier value between the metrics. Hence, dark red equals high $R^2$ of $\geq 0.8$, light red equals $\geq 0.6-0.8$, light blue equals $\geq 0.4-0.6$, medium blue equals $\geq 0.2-0.4$ and dark blue equals $0-0.2$. Metric definitions: Mean (Mean glucose value), Median (Median glucose value), Std (Standard deviation of glucose values), CV (Coefficient of variation), IQR (Interquartile range), Pctile75 (75th percentile of glucose values), Pctile25 (25th percentile of glucose values), TIR (Time in range: 70–180 mg/dL), TITR (Time in tight range: 70–140 mg/dL), TBR1 (Time below range: 54–70 mg/dL), TBR2 (Time below range: <54 mg/dL), TBR (Total time below range: TBR1 + TBR2), TAR1 (Time above range: 180–250 mg/dL), TAR2 (Time above range: >250 mg/dL), TAR (Total time above range: TAR1 + TAR2), Hypo_episodes_n (Number of hypoglycemia events: <70 mg/dL), GRI_Hypo (Glucose Risk Index for hypoglycemia), GRI_Hyper (Glucose Risk Index for hyperglycemia), GRI (Glucose Risk Index), CONGA_1H (Continuous Overall Net Glycemic Action over 1 h), CONGA_2H (Continuous Overall Net Glycemic Action over 2 h), CONGA_6H (Continuous Overall Net Glycemic Action over 6 h), CONGA_24H (Continuous Overall Net Glycemic Action over 24 h), MAGE (Mean Amplitude of Glycemic Excursion), Mobility (Signal mobility), DTpM (Distance traveled per minute), FGxP (Fasting glucose proxy), GMI (Glucose Management).

comprehensive set of metrics that capture key characteristics of glucose dynamics. The widespread adoption of this tool across various research populations could facilitate the identification of CGM metrics with the greatest clinical relevance in different contexts and patient groups. This can enhance the efficient and effective use of CGM in clinical research and practice. Additionally, a case study analyzing the coefficient of determination ($R^2$)

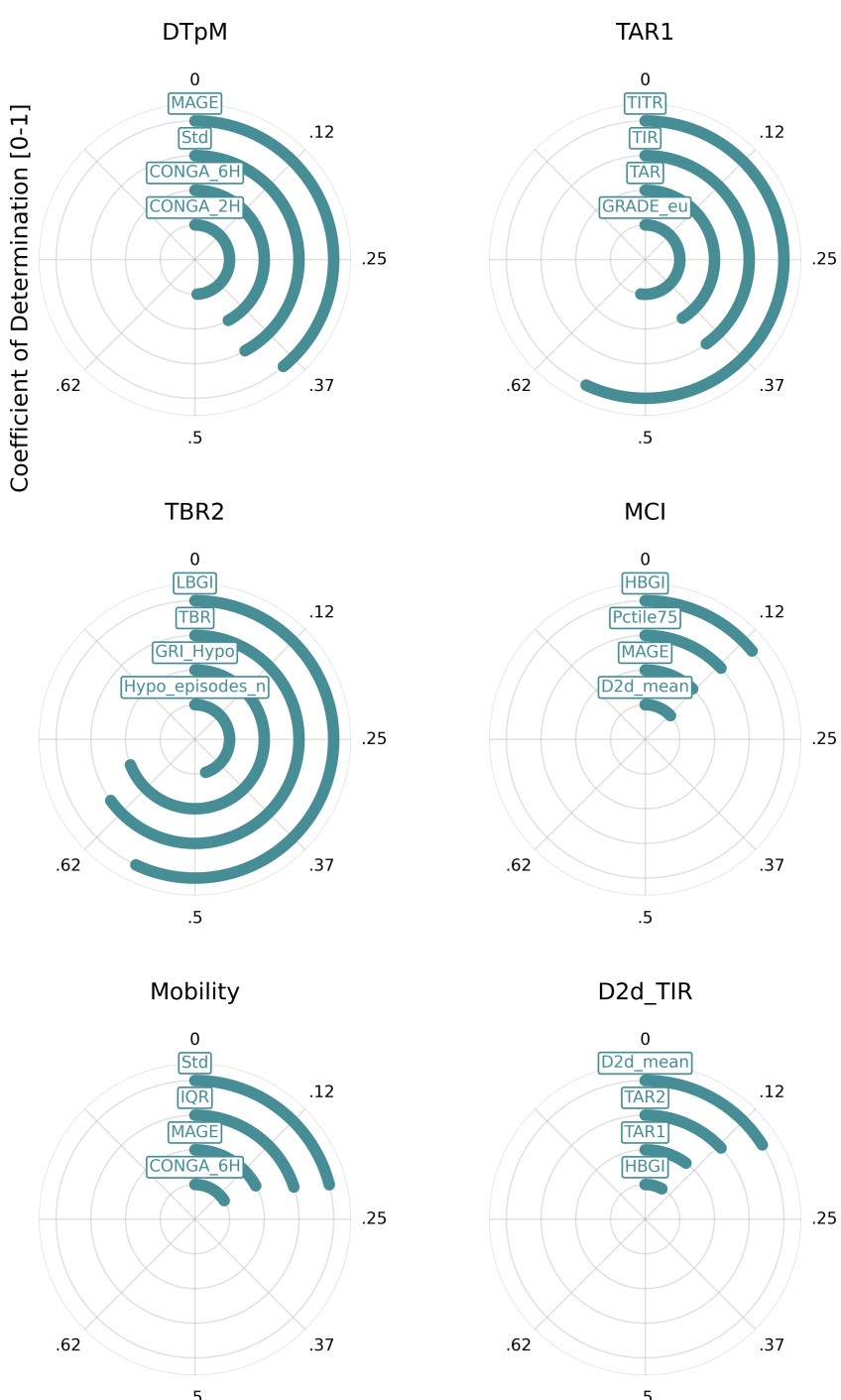

**Figure 5** **Metrics with the lowest cumulative $R^2$ values.** This figure illustrates the six metrics with the lowest cumulative $R^2$ values across all metrics. For each of these metrics, the corresponding highest $R^2$ values are depicted in a circular plot. Metric definitions: DTpM (Distance traveled per minute), TAR1 (Time above range: 180–250 mg/dL), TBR2 (Time below range: <54 mg/dL), MCI (Multiscale Complexity Index), Mobility (Signal mobility), D2d_TIR (Day-to-day standard deviation of time in range).

between the derived CGM metrics in people with insulin-treated T2DM demonstrated that different metrics capture distinct aspects of glucose dynamics. Notably, certain metrics exhibit a low $R^2$ with all other metrics, suggesting that they capture unique, independent information. Further research is required to explore the clinical implications of these findings. Moreover, a direct comparison between QoCGM and established CGM analysis tools in Python and R could provide additional insights into the distinct advantages and potential discrepancies across platforms. While such a comparative evaluation is beyond the scope of this paper, it represents an important direction for understanding QoCGM's added value in CGM data analysis and presentation.

### Funding
The study on the QoCGM development was funded by i-SENS, Inc (Seoul, South Korea). The funders had no role in study design, data collection and analysis, decision to publish, or preparation of the manuscript.

### Grant Disclosures
The following grant information was disclosed by the authors:
i-SENS, Inc (Seoul, South Korea).

### Competing Interests
Simon Cichosz received funding from i-SENS, Inc (Seoul, South Korea) and Simon Cichosz's involvement with the company did not influence the design, implementation, or interpretation of the study. The study was conducted independently, and the authors declare that their involvement with i-SENS, Inc (Seoul, South Korea) did not influence the findings or conclusions of the study. Morten Hasselstrom Jensen has received consultant fees from Abbott.

### Author Contributions
- Simon Cichosz conceived and designed the experiments, performed the experiments, analyzed the data, prepared figures and/or tables, authored or reviewed drafts of the article, and approved the final draft.
- Stine Hangaard conceived and designed the experiments, authored or reviewed drafts of the article, and approved the final draft.
- Thomas Kronborg conceived and designed the experiments, authored or reviewed drafts of the article, and approved the final draft.
- Peter Vestergaard conceived and designed the experiments, authored or reviewed drafts of the article, and approved the final draft.
- Morten Hasselstrøm Jensen conceived and designed the experiments, authored or reviewed drafts of the article, and approved the final draft.

### Human Ethics
The following information was supplied relating to ethical approvals (i.e., approving body and any reference numbers):

Ethical Approval: DiaMonT trial was approved by the Regional Ethical Committee of North Jutland, Denmark (N-20200068).

## Data Availability

The individual data (CGM metrics) are available in the Supplemental File.

The code is available at GitHub and Zenodo:

- https://github.com/simcich/QoCGM

- simcich. (2025). simcich/QoCGM: v-1.01-beta (v1.0.1). Zenodo. https://doi.org/10.5281/zenodo.15018569.

## Supplemental Information

Supplemental information for this article can be found online at http://dx.doi.org/10.7717/peerj.19501#supplemental-information.

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
