# Peer review of "From data to insights: a tool for comprehensive Quantification of Continuous Glucose Monitoring (QoCGM)"

_PeerJ, doi:10.7717/peerj.19501_

## Round 0.1 · original submission · Major Revisions

Dear Dr. Shaik Cichosz,

Your manuscript entitled “From data to insights: a tool for comprehensive Quantification of Continuous Glucose Monitoring (QoCGM)", which you submitted to PeerJ, has been reviewed by the editor and 3 external reviewers.

The reviewers generally support your work but have raised significant concerns that must be addressed before the manuscript can move forward. I would happily reconsider your manuscript if you undertake these substantial revisions and resubmit.

If you decide to resubmit the revised version, please summarize all the improvements made in the new version and give answers to all critical points raised in the reviewers’ report in an accompanying letter. Copy and paste each and every reviewer's comment above your response.

Please note that resubmitting your manuscript does not guarantee eventual acceptance. Since the requested changes are major, the revised manuscript will undergo a second round of review by the same reviewers. I must emphasize that the acceptability of the revision will depend upon the resolution of the points raised by the reviewers.

Sincerely yours,
Stefano Menini

Reviewer 1 ·

Basic reporting

-

Experimental design

-

Validity of the findings

-

Additional comments

The paper introduces QoCGM, an open-source tool for analyzing Continuous Glucose Monitoring (CGM) data in MATLAB. The authors highlight that current tools available in R and Python, such as cgmquantify, iglu, and CGManalyzer, do not fully meet all analytical needs, especially for clinically relevant metrics and MATLAB users. QoCGM is designed to preprocess CGM data, handle missing values, and compute a wide range of metrics covering glycemic control, variability, risk assessment, and complexity analysis.

Some questions:

How does QoCGM handle sensor calibration errors, especially those occurring at the beginning or end of CGM data collection?

Could you clarify whether QoCGM applies any smoothing techniques beyond interpolation, particularly for noisy CGM signals that could affect variability calculations?

How does QoCGM compare in performance (speed and memory efficiency) with existing tools in R and Python, given MATLAB’s computational constraints?

Does the software allow real-time CGM data processing, or is it strictly limited to post-hoc analysis on stored datasets?

In the case study, were there any significant differences in metric correlation between daytime and nighttime periods? If so, how does QoCGM account for these differences?

Have you considered adding machine learning-based anomaly detection for unexpected glucose fluctuations instead of relying on predefined thresholds?

What is the justification for using Piecewise Cubic Hermite Interpolating Polynomial (PCHIP) over other interpolation methods, such as spline or linear regression?

Could QoCGM be adapted for integration into clinical decision-making software, or does its output require additional statistical processing before being clinically useful?

Reviewer 2 ·

Basic reporting

The manuscript is generally well-written and adheres to professional standards of English. The language is clear and technically accurate.

The literature review is thorough, and relevant references are appropriately cited. The background provides sufficient context for the study, demonstrating how QoCGM contributes to the existing body of CGM analysis tools. However, the rationale behind the inclusion of novel metrics (e.g., MCI, Mobility, DTpM) could be more explicitly tied to prior research in glucose variability analysis.

The manuscript follows a logical structure, aligning well with standard scientific article formatting. Figures and tables are generally relevant and informative, though Fig. 4 and Fig. 5 would benefit from improved font size and color contrast to enhance readability. Additionally, some figures (e.g., Fig. 2 and Fig. 4) lack clear axis labels and legends. Adding explicit descriptions for each metric would enhance readability.

The raw data availability is acknowledged, and the study appears to adhere to appropriate data-sharing standards. However, clearer documentation or supplementary materials detailing the specific preprocessing steps (e.g., interpolation method validation) would enhance transparency and reproducibility.

Overall, the manuscript meets the basic reporting criteria but would benefit from concise language adjustments, enhanced figure clarity, and broader statistical summaries of the dataset.

Experimental design

The research question is well-defined and relevant, addressing the need for a MATLAB-based CGM analysis tool that extends beyond conventional methods. The study clearly identifies the knowledge gap, highlighting the limitations of existing Python and R-based tools and proposing QoCGM as a comprehensive alternative.

The investigation appears to have been conducted rigorously, adhering to high technical and ethical standards. The use of CGM data from the DiaMonT trial (NCT04981808) ensures that the dataset is clinically relevant. However, further clarification on participant selection criteria, sensor types, and measurement conditions would strengthen the methodological transparency. For example, details on data exclusion criteria (e.g., missing values threshold), variations in CGM device accuracy, or preprocessing decisions affecting glucose variability metrics should be explicitly described.

The methodology is generally well-documented, with step-by-step explanations of QoCGM’s computational pipeline. However, more information is needed on the validation of interpolation methods (e.g., PCHIP vs. linear or spline interpolation) to assess potential biases introduced during missing data handling.

Overall, the study meets the journal's standards for experimental design.

Validity of the findings

The manuscript presents a well-structured analysis with a clear methodology, and the underlying data appears robust and statistically sound. The study effectively demonstrates the utility of QoCGM for analyzing CGM data, and conclusions are generally well-supported by the presented results.
However, certain aspects require further validation to strengthen the reliability of the findings:

1. While QoCGM introduces new features and metrics, the manuscript does not sufficiently compare its output with established CGM analysis tools, for example, those implemented in Python and R. A direct comparison between QoCGM and these tools using the same dataset would be valuable to assess differences in computational outputs and potential discrepancies in metric calculations. Additionally, beyond standard CGM metrics, QoCGM includes visualization functionalities such as those demonstrated in Fig.1 and Fig.2. It would be beneficial to compare these visualization outputs with equivalent figures generated using Python- or R-based CGM analysis tools. This would help determine whether QoCGM provides distinct advantages in data presentation or whether similar insights could be obtained using alternative methods.

2. The manuscript primarily presents individual case studies (e.g., Fig. 1 and Fig. 2) to illustrate QoCGM’s functionalities. However, it lacks a comprehensive summary analysis of the full dataset (n = 331). While some descriptive statistics are provided, it would be beneficial to include aggregate analyses that summarize the distributions of key CGM metrics across the entire cohort.

·

Basic reporting

The article is very well written, using clear, unambiguous English throughout.

The article provides a good description of the background and rationale for developing the QoCGM open-source tool.

Two relevant references appear to have been missed.
1) “Continuous glucose monitoring and metrics for clinical trials: an international consensus statement” by Battelino et al (doi: 10.1016/S2213-8587(22)00319-9). This article outlines the recommended CGM metrics for reporting in clinical research studies.

2) “Statistical Packages and Algorithms for the Analysis of Continuous Glucose Monitoring Data: A Systematic Review” by Olsen et al (doi: 10.1177/19322968231221803). This systematic review provides an excellent overview of 23 statistical algorithms/analytical tools that exist for the calculation of CGM metrics, and the capabilities of each tool. The review highlights that there are three packages available for MATLAB (AGATA, MAGECAA, and GVAP). In particular, the open-source AGATA package (https://github.com/gcappon/agata, https://gcappon.github.io/agata/) appears to calculate a comprehensive collection of metrics.

As the article describes the implementation of a new analytical tool, it does not follow the ‘standard sections’ for a research article. However, the structure presented is logical and provides clarity to the text.

Figures are clear and of sufficient resolution.

The supplementary raw data contains the summary metrics produced by the QoCGM analytical tool for the 324 participants study sample (DiaMonT trial).

Experimental design

The article presents a Bioinformatics Software Tool and states the rationale for developing the tool, which is within the Aims and Scope of the journal.

As mentioned above, the assertion that 'there is a notable gap in the availability of updated, open-source tools for the MATLAB environment' is not wholly true given the existence of the AGATA package. However, the package may indeed provide some 'newer' metrics that the AGATA package does not, and out-of-the-box 'offers features for examining night/day 72 glucose variations and day-to-day changes'.

On the whole, the authors have provided detailed and clear instructions on the implementation and design decisions for the QoCGM analytical tool. I have a few questions:

Lines 88-89: Sampling tolerance. How are measurements that fall outside the sampling tolerance threshold ‘corrected’, specifically those which occur too early? Are they considered duplicates and removed?

Line 95-96: It wasn’t immediately clear to me what figures are plotted. Is Figure 1 an example of the kind of figure produced when the plotting flag is set to 1? If not, can an example plot be included as supplementary?

Lines 261-262: Why are hypoglycaemic events/episodes reported, but not hyperglycaemic events/episodes? Hyperglycaemic episodes are listed as a key CGM metric in the Danne et al 2017 paper and the Battelino 2022 paper.

Validity of the findings

The details regarding the intended purpose and calculation of each of the metrics has been very well presented. However, I was left a little uncertain as to the actual utility of the different metrics in clinical decision making or research, particularly for those metrics not included in consensus statements. Perhaps this is beyond the scope of this paper.

There appear to be inaccuracies in the supplementary raw data file:
- Table 1: Suggests that the first metric recorded should be ‘MonitoringDays’, yet this metric appears to be missing.
- Some of the entries are recorded as having a completeness Rate of (marginally) greater than 100%. How is this possible if duplicates have been removed?
- The values in the columns 'CONGA_1H' through to 'D2d_TIR' appear to be incorrect. For example, for the first row, the CONGA_1H value is given as ‘155.275.245.730.565’? This needs to be corrected.

The conclusions are generally well stated.

---

## Round 0.2 · accepted · Accept

Dear Dr. Cichosz,

Thank you for submitting the revised version of your manuscript. After a thorough review of the changes by the reviewers and me, I am pleased to inform you that all the reviewers' comments have been adequately addressed. Therefore, your manuscript is ready for publication in PeerJ.

Sincerely yours,

Stefano Menini

Reviewer 1 ·

Basic reporting

The authors have made a great effort to improve the quality of the paper, which in my opinion, would be suitable for publication.

Experimental design

Experimental design is rigorous.

Validity of the findings

Results are robust and consistent.

Additional comments

The authors have diligently followed the recommendations of the reviewers.

Reviewer 2 ·

Basic reporting

The authors have fully addressed the review comments, and the manuscript is now recommended to be published without further revisions.

Experimental design

Sufficient for PeerJ

Validity of the findings

Sufficient for PeerJ